# Treatment of Mouse Sperm with a Non-Catalytic Mutant of PLA2G10 Reveals That PLA2G10 Improves In Vitro Fertilization through Both Its Enzymatic Activity and as Ligand of PLA2R1

**DOI:** 10.3390/ijms23148033

**Published:** 2022-07-21

**Authors:** Roland Abi Nahed, Magali Dhellemmes, Christine Payré, Emilie Le Blévec, Jean-Philippe Perrier, Sylviane Hennebicq, Jessica Escoffier, Pierre F. Ray, Corinne Loeuillet, Gérard Lambeau, Christophe Arnoult

**Affiliations:** 1Université Grenoble Alpes, F-38000 Grenoble, France; rolandabinahed@gmail.com (R.A.N.); magali.dhellemmes@univ-grenoble-alpes.fr (M.D.); emilie.leblevec@gmail.com (E.L.B.); christophe.arnoult@cnrs.fr (J.-P.P.); shennebicq@chu-grenoble.fr (S.H.); jessica.escoffier@univ-grenoble-alpes.fr (J.E.); pray@chu-grenoble.fr (P.F.R.); corinne.loeuillet@univ-grenoble-alpes.fr (C.L.); 2Institute for Advanced Bioscience, INSERM 1209, CNRS UMR 5309, F-38700 La Tronche, France; 3Institut de Pharmacologie Moléculaire et Cellulaire, Université Côte d’Azur, CNRS, Valbonne Sophia Antipolis, F-06560 Valbonne, France; payre@ipmc.cnrs.fr (C.P.); lambeau@ipmc.cnrs.fr (G.L.); 4CHU de Grenoble, Centre d’AMP-CECOS, CS1021, F-38000 Grenoble, France; 5CHU de Grenoble, UF de Biochimie et Génétique Moléculaire, F-38000 Grenoble, France

**Keywords:** in vitro fertilization, sperm, capacitation, acrosome reaction, PLA2G10, PLA2R1

## Abstract

The group X secreted phospholipase A2 (PLA2G10) is present at high levels in mouse sperm acrosome. The enzyme is secreted during capacitation and amplifies the acrosome reaction and its own secretion via an autocrine loop. PLA2G10 also improves the rate of fertilization. In in vitro fertilization (IVF) experiments, sperm from *Pla2g10*-deficient mice produces fewer two-cell embryos, and the absence of PLA2G10 is rescued by adding recombinant enzymes. Moreover, wild-type (WT) sperm treated with recombinant PLA2G10 produces more two-cell embryos. The effects of PLA2G10 on mouse fertility are inhibited by sPLA_2_ inhibitors and rescued by products of the enzymatic reaction such as free fatty acids, suggesting a role of catalytic activity. However, PLA2G10 also binds to mouse PLA2R1, which may play a role in fertility. To determine the relative contribution of enzymatic activity and PLA2R1 binding in the profertility effect of PLA2G10, we tested H48Q-PLA2G10, a catalytically-inactive mutant of PLA2G10 with low enzymatic activity but high binding properties to PLA2R1. Its effect was tested in various mouse strains, including *Pla2r1*-deficient mice. H48Q-PLA2G10 did not trigger the acrosome reaction but was as potent as WT-PLA2G10 to improve IVF in inbred C57Bl/6 mice; however, this was not the case in OF1 outbred mice. Using gametes from these mouse strains, the effect of H48Q-PLA2G10 appeared dependent on both spermatozoa and oocytes. Moreover, sperm from C57Bl/6 *Pla2r1*-deficient mice were less fertile and lowered the profertility effects of H48Q-PLA2G10, which were completely suppressed when sperm and oocytes were collected from *Pla2r1*-deficient mice. Conversely, the effect of WT-PLA2G10 was not or less sensitive to the absence of PLA2R1, suggesting that the effect of PLA2G10 is polymodal and complex, acting both as an enzyme and a ligand of PLA2R1. This study shows that the action of PLA2G10 on gametes is complex and can simultaneously activate the catalytic pathway and the PLA2R1-dependent receptor pathway. This work also shows for the first time that PLA2G10 binding to gametes’ PLA2R1 participates in fertilization optimization.

## 1. Introduction

The superfamily of mammalian phospholipases A2 (PLA_2_s) contains more than 30 enzymes and is divided into intracellular PLA_2_s (cPLA2s and iPLA2s) and secreted PLA_2_s (sPLA2s) [1]. Among the 12 mouse sPLA_2_ genes [2], the group X sPLA_2_ (PLA2G10) has unique enzymatic properties, with a high affinity for arachidonic acid-containing phospholipids [3], and is highly expressed in testis [4]. Its absence in sperm cells leads to a decrease in male fertility, with KO males producing a reduced number of pups [5], suggesting that sperm PLA2G10 is necessary for optimal sperm physiology. We confirmed this hypothesis by showing that the yield of two-cell embryos generated by in vitro fertilization (IVF) using sperm collected from *Pla2g10*-deficient males was strongly decreased and rescued by adding recombinant PLA2G10 in the medium. The absence of PLA2G10 in sperm impacted IVF more severely than natural mating, suggesting a compensating sPLA_2_ activity from the female tract. PLA2G10 is located in the acrosome and secreted during the acrosome reaction [5]. During in vitro mouse sperm capacitation, a maturation step necessary for the acrosome reaction, a significant proportion of sperm performs a spontaneous early acrosome reaction [5], and enzymes released from these sperm modify the lipid composition of the capacitating sperm. Thus, at the cellular level, the enzyme participates in sperm membrane lipid changes induced by capacitation and preferentially cleaves DocosaHexaenoic Acid (DHA)- or DocosaPentaenoic Acid (DPA)-containing phospholipids in sperm [6]. The production of fatty acids is also necessary for the conformational change in syntaxin [7], a protein involved in the acrosome reaction, and PLA2G10 amplifies their production, increasing the level of acrosome-reacted sperm [8]. Interestingly, adding recombinant PLA2G10 in the medium during capacitation of WT sperm leads to an increase in two-cell embryos produced by IVF [5,9,10]. Similar results were obtained with bovine sperm treated with mouse PLA2G10 [11], showing the interest in this enzyme as a drug to improve fertility. Despite PLA2G10 involvement in the acrosome reaction [8], we demonstrated that the effect of PLA2G10 on fertilization outcome was independent of the level of acrosome reaction and was not mimicked by molecules triggering acrosome reactions such as progesterone. *PLA2G10-dependent improvement of fertilization* is remarkable, and its action is unique since all other tested sPLA_2_s were unable to mimic its positive effect on fertilization outcome [9], whereas they trigger acrosome reaction. However, the precise molecular mechanism triggered by PLA2G10 to improve fertilization is not identified so far. Based on the molecular properties of PLA2G10, two mechanisms can be proposed. First, PLA2G10, as a hydrolytic enzyme, cleaves glycerophospholipids to release various fatty acids and lysophospholipids. These lipid metabolites can activate numerous cellular pathways in different tissues directly or indirectly, including reproductive organs [12]. In particular, these lipids are known to promote the acrosome reaction and the fusion of gametes during fertilization [8,13,14,15]. As such, these compounds may be responsible for the *PLA2G10-dependent improvement of fertilization*. Alternatively, PLA2G10 could induce some biological responses through its ability to bind to specific membrane receptors. There are indeed more and more reports showing that sPLA_2_s are ligands for different protein families located at the plasma membrane including PLA2R1 [16], integrins [17], heparan sulfate proteoglycans [18,19], and vascular endothelial growth factor receptors (VEGFRs) [20]; an updated list of sPLA_2_ ligands was recently reviewed by [21]. In particular, the biological response of PLA2G10 may thus be related to its binding property to PLA2R1, to which mouse PLA2G10 binds with a nanomolar affinity [16,22]. However, the presence of sPLA_2_ binding proteins in gametes and their possible physiological function were never assessed, and the aim of this work was to address this question. To tackle this question, we used a defective mutant of PLA2G10, devoid of enzymatic activity (this mutant has less than 0.1% of WT enzymatic activity) but retaining PLA2R1 binding activity. This tool is the H48Q active site mutant of PLA2G10 [23,24], where the catalytic histidine 48 is replaced by glutamine (named H48Q-PLA2G10 in this report). Any pharmacological effect of H48Q-PLA2G10 would be an indication that WT-PLA2G10 can participate in physiological processes not just as enzyme but also as ligand for a receptor.

## 2. Results

### 2.1. H48Q-PLA2G10 Has No Effect on the Acrosome Reaction

The acrosome reaction is very sensitive to sPLA_2_ enzymatic activity, as we have shown previously that nM concentrations of PLA2G10 induce a significant increase in acrosome reaction [5,9]. The rates of sPLA_2_-induced acrosome reaction of C57BL/6 capacitated sperm incubated with WT-PLA2G10 or H48Q-PLA2G10 were compared (Figure 1). WT-PLA2G10 induced a strong increase in acrosome reaction, whereas the mutant was completely ineffective at increasing the acrosome reaction (Figure 1). The absence of effect of H48Q-PLA2G10 suggests that acrosome reaction is not controlled by PLA2G10-binding proteins and confirms the importance of fatty acids in triggering an acrosome reaction. Moreover, this result indicates that an acrosome reaction is a good index of sPLA_2_ enzymatic activity and confirms that the H48Q-PLA2G10 mutant has no catalytic activity and could be used as a tool to detect the presence of sPLA_2_ binding proteins.

### 2.2. H48Q-PLA2G10 Has No Effect on Parthenogenesis

We also wanted to assess a possible action of H48Q-PLA2G10 on unfertilized oocytes. It is indeed important, before assessing a possible role of PLA2G10-binding proteins in fertilization, to assess a possible effect of WT- and H48Q-PLA2G10 on parthenogenesis. To test this hypothesis, we incubated C57BL/6 non-fertilized MII oocytes obtained by hormonal stimulation with WT-PLA2G10 or H48Q-PLA2G10 at different concentrations (Figure 2). We tested concentrations between 0.2 and 20 nM because oocytes in the IVF experiments were never exposed to a concentration above 2 nM of sPLA_2_ (see materials and methods section). Using a one-way ANOVA test, no significant statistical difference (*p* > 0.05) was observed between the mean number of two-cell embryos obtained after incubation of MII oocytes in M16 medium only or with different concentrations of WT-PLA2G10 and of H48Q-PLA2G10. These results demonstrate that neither WT-PLA2G10 nor H48Q-PLA2G10 were able to induce parthenogenic activation. In other words, neither enzymatic activity nor ligand activity of PLA2G10 are sufficient to trigger parthenogenesis and alter the results of fertilization experiments.

### 2.3. H48Q-PLA2G10 Improves IVF Yield

We have previously shown that addition of recombinant WT-PLA2G10 during mouse sperm capacitation improves the yield of two-cell embryos in IVF experiments [5,9,10]. We call herein this observable fact “sPLA_2_-dependent improvement of fertilization”. Although we have shown that the PLA2G10 effect is inhibited by a specific small molecule sPLA_2_ inhibitor (LY329722), the precise mode of action of WT-PLA2G10 has remained elusive so far and might be due to either the production of lipid metabolites or the interaction of PLA2G10 with a sPLA_2_ receptor. In the first set of experiments, we used male and female gametes from the C57BL/6J strain and tested the pharmacological effect of H48Q-PLA2G10. Interestingly, H48Q-PLA2G10 was as efficient as WT-PLA2G10 at improving the yield of two-cell embryos (Figure 3), suggesting that sPLA2 binding proteins may be involved in the optimization process.

### 2.4. “sPLA_2_-Dependent Improvement of Fertilization” Induced by H48Q-PLA2G10 Is Strain Dependent

The discovery of “sPLA_2_-dependent optimization of fertilization” was made with OF1 outbred mice, which are quite different from the classical inbred C57BL/6J mice [5]. We asked whether we could reproduce the effect of H48Q-PLA2G10 on IVF with gametes from OF1 mice. Surprisingly, treating OF1 sperm with H48Q-PLA2G10 did not improve the rate of two-cell embryos, while WT-PLA2G10 did (Figure 4A). This suggests that the target of H48Q-PLA2G10 or a component of its mechanism of action is absent in the OF1 strain. This difference did not appear to be due to a different sensibility of OF1 sperm to the enzymatic activity of WT-PLA2G10 and its inactive mutant, as the acrosome reaction of OF1 sperm was sensitive to WT-PLA2G10 but not to H48Q-PLA2G10, as for C57Bl/6J sperm (Figure 4B). It is worth noting that treating sperm with the catalytically-active WT-PLA2G10, as previously demonstrated, boosted the IVF reaction. These results suggest a more complex than expected mode of action of PLA2G10, which may be polymodal with two superimposed pathways: a first one that appears to be dependent on catalytic activity and more prominent in OF1 mice, and a second one that would be independent of enzymatic activity and more prominent in C57BL/6J mice.

To better analyze the difference between the two strains, we performed inter-strain IVF between OF1 and C57BL/6 gametes. We first mixed sperm from OF1 males, treated or not with H48Q-PLA2G10, with MII oocytes from either C57BL/6 or OF1 females. Figure 5 shows that switching the origin of oocytes from OF1 to C57BL/6 is sufficient to restore the boosting effect of H48Q-PLA2G10 on fertilization outcome (Figure 5A,B). Figure 5C compares the percentage of IVF increase (calculated as “(rate of fertilization with sPLA_2_-treated sperm ×100/rate of fertilization with control sperm)” ×100) obtained in both experimental conditions with H48Q-PLA2G10 and clearly shows the restoration of the positive effect of treatment of sperm by H48Q-PLA2G10 when oocytes were obtained from C57BL/6 females. This result demonstrates that the IVF effect of H48Q-PLA2G10 is at least partially mediated by the C57BL/6 female gametes.

We next performed the mirror experiment, mixing sperm from C57BL/6 males, treated or not with H48Q-PLA2G10, with oocytes from either C57BL/6 or OF1 females. With oocytes from both strains, treatment of sperm with H48Q-PLA2G10 led to an increase in fertilization (Figure 6A,B). As in Figure 5C, Figure 6C compares the percentage of IVF increase obtained in both strain conditions after treatment with H48Q-PLA2G10 versus untreated sperm. This result first shows that switching the origin of sperm from OF1 to C57BL/6 restores the effect of H48Q-PLA2G10 on fertilization (Figure 4A vs. Figure 6B). Nevertheless, when OF1 oocytes were used, the boosting effect of H48Q-PLA2G10 was significantly weaker than with C57BL/6 oocytes (Figure 6B), confirming that oocytes play the most important role in the profertility effect of H48Q-PLA2G10 on sperm.

### 2.5. Expression of PLA2R1 in Gametes and Role in IVF

In the mouse species, PLA2G10 was identified as an endogenous ligand for PLA2R1, with an affinity in the low nM range [16,22,25]. However, the expression of PLA2R1 in gametes has not been the subject of specific studies and its possible role in reproduction is unknown. *Pla2r1*-deficient mice originally produced by Hanasaki et al. have been found to be viable and fertile [26]. In accordance, we found that our line of *Pla2r1*-deficient is also normally fertile, even when *Pla2r1*-deficient male mice are crossed with *Pla2r1*-deficient females. These breedings produced litters of size and sex similar to that of wild-type siblings (data not shown). According to the Genevestigator database (Affymetrix studies), mRNA expression levels of *Pla2r1* in mouse reproductive tissues and gametes are in the medium range, comparable to expression in other tissues, with, for instance, medium to high expression levels in liver and bladder (Figure 7A). The receptor is expressed in all spermatogenic cells from spermatogonia to spermatid and in oocytes at similar levels (Figure 7B). In agreement with these data, we confirmed by RT-qPCR a fairly high expression of *Pla2r1* in testis and oocytes, at levels similar to those observed in liver and bladder (Figure 7C).

To assess the role of PLA2R1 in sperm fertility, we first performed IVF experiments with sperm from *Pla2r1*-deficient mice and oocytes from WT C57BL/6 females. The number of two-cell embryos was lowered from 50.0% to 41.6% (*n* = 5, *p* < 0.0001) when IVF was performed with sperm from WT littermates versus *Pla2r1*-deficient mice, suggesting a role of PLA2R1 in spermatogenesis or sperm physiology (Figure 8).

### 2.6. H48Q-PLA2G10-Induced Improvement of Fertilization Is Altered with Gametes from Pla2r1 -Deficient Mice

To evaluate a possible role of PLA2R1 in the improvement of fertilization, we next compared the percentage of increase in IVF after addition of sPLA_2_ when IVF is carried out with sperm from WT or *Pla2r1*-KO males and treated with WT- or H48Q-PLA2G10 (Figure 9). Treatment of sperm from WT or *Pla2r1*-KO males with WT-PLA2G10 produced the same improvement of IVF (24.2 ± 3.3% (*n* = 6) versus 27.0 ± 3.6% (*n* = 5), *p* = 0.57, Figure 9A), suggesting that PLA2R1 expression in male mice is dispensable for the IVF effect of WT-PLA2G10. As shown in Figure 3, the fertilization rate was improved by H48Q-PLA2G10 when IVF experiments were performed with WT sperm (30.6 ± 2.7% (*n* = 13)). However, the fertilization rate induced by H48Q-PLA2G10 was significantly decreased when IVF experiments were performed with sperm from *Pla2r1*-deficient mice (17.0 ± 4.3% (*n* = 5), *p* = 0.017, Figure 9B). It is worth noting that the effect of H48Q-PLA2G10 was not fully abolished (it remained 17.0%) when *Pla2r1*-KO sperm were used (Figure 9B), again suggesting that H48Q-PLA2G10 may not only stimulate sperm but also oocytes, in line with the importance of oocytes in H48Q-PLA2G10 effects (Figure 5 and Figure 6) and the expression of PLA2R1 in oocytes (Figure 7). To challenge this hypothesis, we performed the same experiments with both sperm and oocytes collected from *Pla2r1*-KO mice (Figure 10). The improvement effect induced by either WT-PLA2G10 or H48Q-PLA2G10 was totally abolished, confirming our hypothesis of the role of PLA2R1 on oocytes as well. Moreover, the rate of two-cell embryos obtained in IVF experiments performed with both sperm and oocytes from *Pla2r1*-KO mice did not further decrease the rate of two-cell embryos in comparison with the rate observed when IVF was made with sperm from *Pla2r1*-KO mice (Figure 8).

Altogether, these results show that (i) PLA2R1 is present in both types of gametes, (ii) its absence in sperm negatively impacts the yield of IVF, without further impact when PLA2R1 is also absent in oocytes and (iii) the H48Q-PLA2G10 effect is strongly affected by the *Pla2r1* genotype of gametes and completely abolished when both gametes were collected from *Pla2r1*-KO animals.

Finally, in an attempt to understand why the OF1 strain is not sensitive to H48Q-PLA2G10, we determined the level of mRNA expression of *Pla2r1* in sperm and oocytes from OF1 versus C57BL/6 mice by RT-qPCR. A significant two-fold lower expression of *Pla2r1* was observed in sperm from OF1 males, whereas a trend but not significant increase in expression was observed in oocytes (Figure 11).

## 3. Discussion

We have previously shown that mouse PLA2G10 is secreted during the acrosome reaction and that it plays two roles, first on the rate of acrosome reaction via an amplification loop [8] and second on the rate of fertilization, where the enzyme improves both reactions [5,9]. Using a catalytically-inactive mutant of PLA2G10, we challenged the possibility that the action of recombinant PLA2G10 on fertility is not solely due to its enzymatic activity, while its effect on acrosome reaction was clearly dependent on enzymatic activity. Overall, we show that the profertility effect of PLA2G10 is polymodal and associated with the catalytic properties of the enzyme but also with a non-catalytic mechanism, likely associated with PLA2R1, with the latter present in both sperm and oocytes. Furthermore, this polymodal effect of recombinant PLA2G10 seems to rely not only on sperm but also on oocytes, highlighting an unexpected complexity in the mode of action of PLA2G10.

### 3.1. Coexistence of Two Modes of Action

For the first time, we challenged the possibility that a catalytically-inactive mutant of PLA2G10 is involved in *PLA2G10-dependent improvement of fertilization* using gametes from three different lineages, which are OF1, C57BL/6 and *Pla2r1*-KO strains. In the OF1 model, whereas the active WT-PLA2G10 enzyme is potent at improving fertilization outcome, the inactive H48Q-PLA2G10 enzyme has no effect. This result indicates that PLA2G10-dependent improvement of fertilization is triggered by the enzymatic activity of WT-PLA2G10 in OF1 mice. Moreover, PLA2G10 kept its ability to improve the fertilization outcome in C57BL/6 and *Pla2r1*-KO mice when sperm from both strains were treated. On the other hand, in the C57BL/6 model, H48Q-PLA2G10 was also potent at boosting the fertilization outcome. Altogether, these results highlight the presence of two pathways, one probably involving release of fatty acids and/or lysophospholipids and another involving a sPLA_2_ binding protein, probably PLA2R1. The seemingly absence of an sPLA2 binding pathway in the OF1 model is not due to the absence of PLA2R1 in oocytes because *Pla2r1* is similarly expressed in both strains. On the other hand, *Pla2r1* is significantly less expressed in OF1 sperm. Nevertheless, it is difficult to assess the impact of this decrease on H48Q-PLA2G10. The differences observed in the effect of PLA2G10 in fertility between OF1 and C57BL/6 mice may also be considered with two other observations in the sPLA2 field. First, it should be highlighted that inbred mice such as C57BL/6 are naturally-deficient for *Pla2g2a*, another important mouse sPLA2 [27]. Conversely, inbred Balb/C or outbred mice such as CD-1 (and likely OF1) are proficient. The role of PLA2G2A in mouse fertility has not been addressed, but the enzyme is present in various male organs [28], suggesting a direct role of PLA2G2A in fertility or an indirect role, acting on the function of PLA2G10 or even PLA2R1, since mouse PLA2G2A binds to mouse PLA2R1 [16]. Second, it has been observed that the effect of PLA2G5, another sPLA2 involved in the release of lipid mediators in various settings, is dependent on the mouse strain, with the phenotype of *Pla2g5*-KO mice observed in the C57BL/6 background but lost or markedly decreased in the Balb/C genetic background [29,30]. Thus, it would be interesting to analyze the phenotype of *Pla2g10*- and *Pla2r1*-deficient mice, and of recombinant WT- and H48Q-PLA2G10, in mice proficient for *Pla2g2a* and/or deficient for *Pla2g5*.

Moreover, we cannot rule out that OF1 mice carry a mutation in the *Pla2r1* gene, and in particular, in the CTLD4–6 region that binds the sPLA_2_ [31]. This result reinforces the hypothesis of a cellular signaling pathway triggered by PLA2R1, which remains to be characterized. Surprisingly, in the C57BL/6 model, WT- and H48Q-PLA2G10 had a similar effect, whereas we could have expected a stronger effect since WT-PLA2G10 acts on both pathways (production of lipid mediators and possible activation of the receptor). This suggests that both stimulations could converge to the same overall signaling pathway.

### 3.2. PLA2R1 of Both Sperm and MII Oocyte Is Involved in the H48Q-PLA2G10-Dependent Improvement of Fertilization

The involvement of PLA2R1 is supported by the following results: (1) H48Q-PLA2G10 was able to improve fertilization outcomes in the C57BL/6 model, and (2) the effect of H48Q-PLA2G10 was significantly decreased when sperm from *Pla2r1*-KO were used. This latter experiment also indicates that sperm is an actor in this process. Moreover, we have shown that this effect is mediated not only by sperm but also by oocytes because of the following results: (1) the effect of H48Q-PLA2G10 was completely lost when both sperm and oocytes from *Pla2r1*-KO mice were used; (2) in IVF experiments using sperm from OF1 males, switching the origin of oocytes from OF1 to C57BL/6 is sufficient to restore the boosting effect of H48Q-PLA2G10 on fertilization outcome; (3) in IVF experiments using sperm from C57BL/6 males, switching the origin of oocytes from C57BL/6 to OF1 led to a significant decrease in the H48Q-PLA2G10 effect. Altogether, these results demonstrate that the impact of *H48Q-PLA2G10 on the improvement of fertilization* depends on both types of gametes and that sPLA_2_-binding protein pathways in sperm and oocytes are additive. What we do not know is how the oocyte PLA2R1 is stimulated by H48Q-PLA2G10. There are two hypotheses: first, H48Q-PLA2G10, bound on sperm, could be delivered to oocytes by an exchange mechanism, which remains to be explored, or second, the washing carried out at the end of sperm treatment did not remove 100% of H48Q-PLA2G10, and sperm-associated H48Q-PLA2G10 could be delivered in the fertilization dish at the time sperm activate oocyte receptors.

### 3.3. The Positive Effect of H48Q-PLA2G10 on Embryo Production Suggests That PLA2R1 Activates an Intracellular Pathway

Although we have demonstrated that PLA2R1 is involved in sperm and oocyte physiology and that the receptor is involved in the mechanism of action of H48Q-PLA2G10, no in vivo fertility defect was reported for *Pla2r1*-KO mice [26], and a limited in vitro defect was observed in this study. The function of PLA2R1 has been the subject of debate for years, and two hypotheses, not necessarily opposed, have been proposed; in the first, the receptor would be involved in the clearance of circulating sPLA_2_ by internalization for degradation in lysosomes, thus stopping the production of lipid mediators by the sPLA2 enzymatic activity [32]. A soluble form of the receptor may also exert a direct inhibition of the sPLA2 in the extracellular milieu [33]. The second hypothesis proposes that the receptor could activate a cellular signaling pathway allowing an adequate answer of the cell to its changing environment [34,35]. However, besides the role of some intracellular motifs involved in its internalization properties, the exact downstream effectors of PLA2R1, which are the binding partners and the cellular signaling pathways that would be activated upon ligand binding, remain enigmatic [36,37]. For instance, PLA2R1 has been reported to be present in different immune cell types as mast cells, and neutrophils and may be involved in some sPLA_2_-induced effects [38,39,40,41]. Several reports demonstrated that PLA2R1 has an important role in regulating tumor-suppressive responses via activation of Janus kinase 2, yet these effects do not seem to be related to its sPLA2 binding properties [42,43]. To summarize, although complex and still poorly understood, there is a body of evidence showing that PLA2R1 plays a role in the activation of intracellular pathways. Furthermore, not all of the biological effects of PLA2R1 may be dependent on sPLA_2_ binding.

Here, we show (i) that the absence of PLA2R1 in sperm decreases the rate of embryos obtained by IVF, (ii) that the boosting effect obtained by H48Q-PLA2G10 is decreased when sperm from *Pla2r1*-KO mice are used, and (iii) that the boosting effect obtained by H48Q-PLA2G10 is suppressed when sperm and oocytes from *Pla2r1*-KO mice are used. The fact that H48Q-PLA2G10, devoid of catalytic property, is able to boost fertilization in a PLA2R1-dependent manner strongly suggests that PLA2R1-dependent intracellular pathways are activated in sperm and/or oocyte, leading to an improved embryo development. Nevertheless, this result does not allow us to exclude that the binding of WT-PLA2G10 to PLA2R1 also has a positive effect through inhibition of its catalytic activity, which was found to be deleterious on zygote or embryo development [11,44].

### 3.4. Importance of the Physiological Mechanisms for the Optimization of Fertilization

We have previously shown that mouse PLA2G10 is secreted during the acrosomal reaction and that it plays a double role, first in an amplification loop of the acrosome reaction [8] and second in the improvement of the fertilization rate [5,9]. We show in this paper that the improvement of fertilization is supported by two signaling pathways, one using the catalytic properties of the enzyme and the other dependent on PLA2R1, which appears to be expressed in both sperm and oocytes. Although we have demonstrated that PLA2R1 is involved in sperm and oocyte physiology and that the receptor is involved in the mechanism of action of the H48Q-PLA2G10 mutant, no in vivo fertility defect was reported for Pla2r1-KO mice [26] and limited in vitro defects from this study. The PLA2G10/PLA2R1 pathway does not seem essential for mouse reproduction. Among actors supporting sperm physiology and fertilization, two types of proteins have been described. Some proteins are essential and their absence in KO models leads to total infertility. For instance, the sodium/proton exchanger SLC9C1 (sNHE) and the soluble adenylyl cyclase (sAC) are essential for capacitation [45], the CATSPER and SLO3 ion channels for sperm motility or the receptors Izumo, JUNO and CD9 for sperm-egg interaction. In contrast, some proteins are present, but their absence leads to mild to severe subfertility or even no phenotype, showing that their presence is not essential. These proteins are involved in the same functions (capacitation, acrosome reaction, fusion, etc.) but are identified as “non-essential actors” of fertilization [46,47]. However, the importance of these proteins might be underestimated because the use of the mouse model to decipher genotype/phenotype relationships has limitation. Indeed, the phenotyping is performed in particular conditions where KO males are mated with WT females, without competition or in breeding conditions masking complex phenotypes. For instance, the function of MAGE cancer testis antigens to protect the male germline is revealed only when males are subjected to an environmental stress [48]. Another example of the limitation of classical reproductive phenotyping has been emphasized in the study on the importance of the PKDREJ protein in sperm capacitation [49]. Another parameter that is rarely taken into account is the role of these non-essential proteins in litter size. This is a fundamental ecological parameter, and each species has an ideal litter size to resist predation pressure while maintaining a balance between the number of animals and the resources available in the environment. Any dysregulation in this balance, even small, can have a strong impact on the ecology of the species and its possible disappearance [50,51]. It is very likely that these so-called “non-essential” proteins actually play a central role in the balance between reproductive efficiency and the conservation of the species. To challenge this hypothesis, however, it would be necessary to take the deficient animals out of the laboratory, which is ecologically very risky. PLA2G10 and PLA2R1 may be important for such ecological adaptation due to their properties at increasing litter size.

## 4. Material and Methods

### 4.1. Animals

All animal procedures were run according to the French guidelines on the use of animals in scientific investigation with the approval of the local Ethics Committee (ComEth Grenoble N° 318, ministry agreement number # 7128 UHTA-U1209-CA). All animals (OF1 and C57BL/6J) were from Charles River laboratories, Ecully, France. *Pla2r1*-KO animals in the C57BL/6J background were described previously [42]. All animals were 5–8 weeks old for females and 2–6 months old for males at the time of experiments.

### 4.2. Chemical Compounds

M2 medium, M16 medium and BSA were purchased from Sigma-Aldrich (St Quentin Fallavier, France). PMSG and HCG were from Intervet (Beaucouzé, France).

### 4.3. Production of Recombinant sPLA_2_s

Pure recombinant mouse PLA2G10 sPLA2s were produced in *E. coli* by in vitro refolding of inclusion body protein and extensive HPLC purification [16,23]. All purified sPLA_2_s were characterized by gel analysis and mass spectrometry, and specific enzymatic activity was measured using radiolabeled *E. coli* membranes as substrate [3,52].

### 4.4. Capacitation

Mouse sperm, obtained by manual trituration of caudae epididymides, were allowed to swim in M2 medium for 10 min. Sperm were capacitated in M16 medium with 2% fatty-acid-free BSA in the presence or absence of WT-PLA2G10 or H48Q-PLA2G10 at 37 °C in a 5% CO_2_ incubator for different times as specified.

### 4.5. AR Assay

After capacitation, sperm were transferred in PBS and fixed with 4% paraformaldehyde solution for 2 min. Sperm were washed (100 mM ammonium acetate, 2 min), wet-mounted on slides, and air-dried. Slides were rinsed with water and stained with Coomassie blue (0.22%) for 2 min and rinsed again. A total of 100–150 sperm cells/slide were scored per each condition.

### 4.6. In Vitro Fertilization (IVF)

Eggs were collected from mature OF1or C57BL/6 females, synchronized with 5 units of pregnant mare serum gonadotrophin (PMSG) and 5 units of human chorionic gonadotrophin (hCG). Sperm were capacitated for 45–55 min in M16 2% BSA (37 °C, 5% CO_2_) as specified. For treatment with recombinant WT-PLA2G10 or H48Q-PLA2G10, sperm were incubated during capacitation for the last 10 min with 200 nM sPLA_2_, a concentration found to be potent for IVF improvement [5]. After treatment, sperm were washed by centrifugation (500 g, 5 min) to remove unbound enzymes, possible lipid metabolites, and all acrosomal compounds released during induced AR. The dilution of former compounds after centrifugation is around 1/10. Finally, washed sperm were introduced into droplets containing oocytes (50 µL into 500 µL), and the final dilution of unbound enzymes, possible lipid metabolites, and all acrosomal compounds released during induced AR is therefore around 1/100. Oocytes were incubated with 2.5 × 10^5^ capacitated sperm/mL (37 °C, 5% CO_2_) in M16 medium, and unbound sperm were washed away after 4 h incubation. Twenty-four hours after fertilization, the different stages, i.e., unfertilized oocytes and 2-cell embryos (as an indication of successful fertilization) were scored.

### 4.7. RT-PCR: Assessment of Pla2r1 Expression

RNAs were extracted with the RNAzol RT reagent (Sigma-Aldrich) following the manufacturer’s recommendations. RNAs were recovered from 2 caudae epididymides, 50 oocytes at the germinal vesicle stage or 20 mg of tissues (liver, kidney, bladder). RNA concentrations were determined by using the Qubit RNA assay kit (ThermoFisher Scientific, Waltham, MA, USA). In total, 40 ng (oocytes and spermatozoa) or 800 ng (tissues) of total RNA were used to perform the RT step using the iScript cDNA synthesis kit (Biorad, Marnes-la-Coquette, France) in a total volume of 20 µL. Gene expression was assessed by qPCR (2 µL of undiluted cDNA, final volume of 20 µL) with TaqMan Gene Expression Assays (ThermoFisher Scientific) for *Pla2r1* (Mm01329147_m1) and *Gapdh* (Mm99999915_g1) as the reference control. PCRs were performed on the CFX96 Real-Time apparatus System (Biorad). The relative change in *Pla2r1* expression was determined using the 2 (-Delta Delta C(T)) method. Experiments were repeated 4 to 5 times.

### 4.8. Statistics

Statistical analyses were performed with SigmaPlot 10.0 (Systat software, Inc., San Jose, CA, USA) or Graphpad prism 6.1, (GraphPad Software, Inc., San Diego, CA, USA). Statistical tests as specified for each figure were used to compare untreated and sPLA_2_-treated sperm. Data represent mean ± SD. Statistical tests with 2-tailed *p* values ≤ 0.05 were considered as significant.

## Figures and Tables

**Figure 1 ijms-23-08033-f001:**
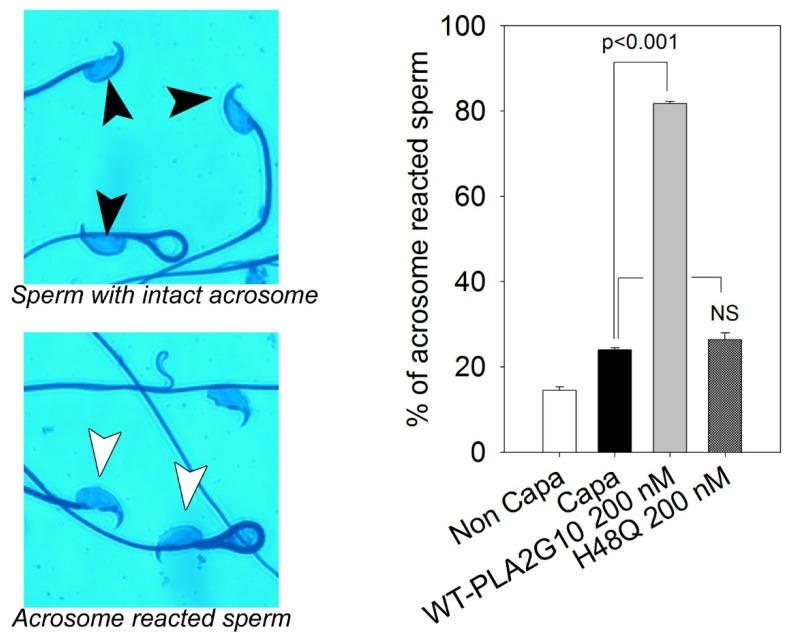
The mutant H48Q-PLA2G10 does not trigger an acrosome reaction contrary to WT-PLA2G10. Sperm from C67BL/6 males were capacitated for 45 min in M16 medium containing 2% BSA, capacitated sperm were treated for 10 min with either 200 nM of WT-PLA2G10 or H48Q-PLA2G10 (noted H48Q), and acrosome reacted sperm were scored using a blue Coomassie technique. In non-acrosome reacted sperm, a dark blue band is present (black arrow head), whereas this band is absent in acrosome reacted sperm (white arrow heads) *n* = 7. n represents the number of biological replicates and for each replicate, more than 100 sperm were assessed per condition. The statistical difference in the mean was assessed using the *t*-test. *p*-value, as indicated. NS, not significantly different.

**Figure 2 ijms-23-08033-f002:**
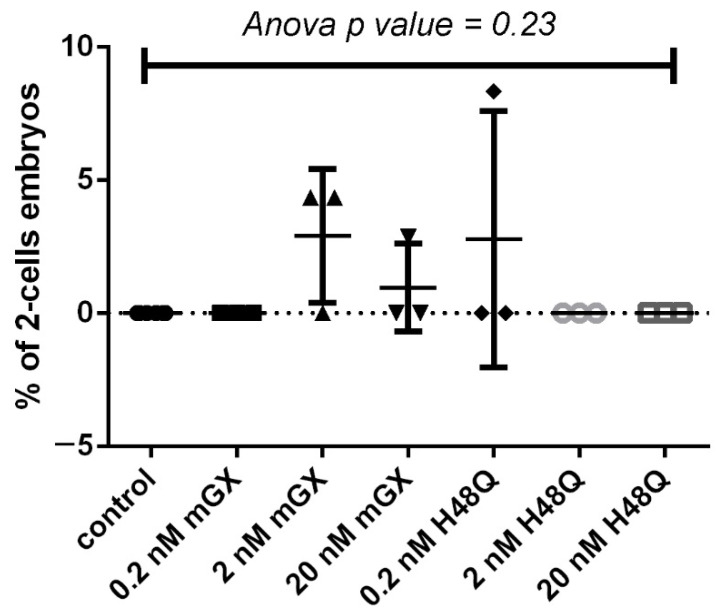
H48Q-PLA2G10 and WT-PLA2G10 do not activate parthenogenesis of MII oocytes from C57BL/6 mice. Batches of MII oocytes (22–36 oocytes per batch) were incubated in M16 medium in an incubator at 37 °C and at 5% CO_2_ 24 h in control medium (black circles) or with different concentrations of WT-PLA2G10 (black squares, up and down triangles) and H48Q-PLA2G10 (noted H48Q) (black diamonds, Grey circles, grey open squares), as indicated. The percentage of pathenogenotes at 2-cell stages was counted after 24 h for each condition. *n* = 3–5, depending on condition. Statistical difference was challenged with a one-way ANOVA test. *p*-value as indicated.

**Figure 3 ijms-23-08033-f003:**
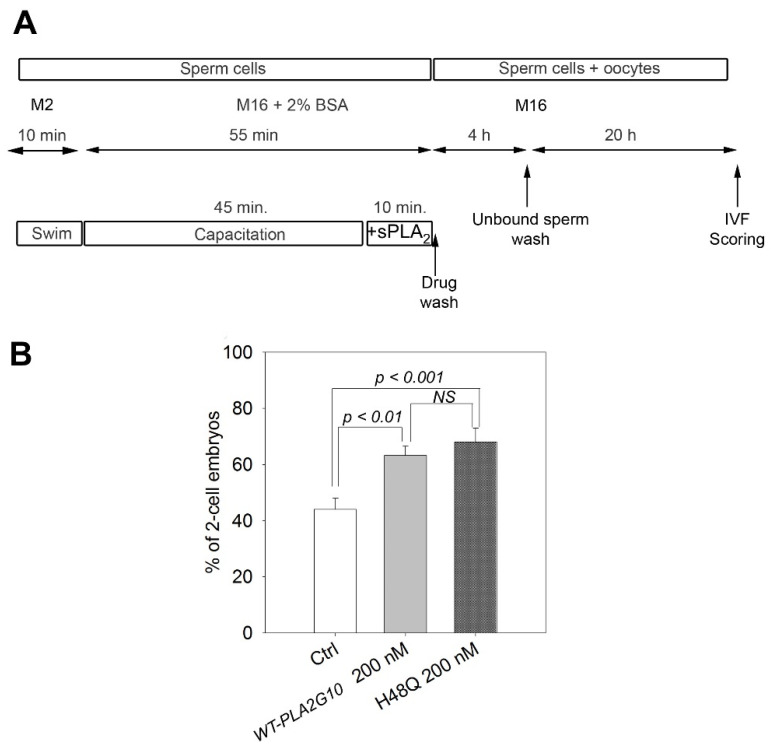
The inactive mutant H48Q-PLA2G10 is as efficient as the catalytic WT-PLA2G10 enzyme in boosting in vitro fertilization of gametes from C57BL/6 mice. (**A**) Schematic drawing of IVF experiments. Sperm were capacitated for a total duration of 55 min in M16-2% BSA and incubated at the end of the capacitation period with WT-PLA2G10 or its catalytic inactive mutant H48Q-PLA2G10 (noted H48Q). After treatment, sperm were washed out by centrifugation to remove unbound drugs, PLA2G10 catalytic products and all acrosomal compounds released during sPLA_2_-induced AR. Finally, washed sperm were introduced into droplets containing MII oocytes (20–109 oocytes per experiment). After 4 h of gamete mixing, unbound sperm were washed away, and IVF outcomes were scored at 24 h. (**B**) Comparison of IVF outcomes obtained with sperm treated with either 200 nM WT-PLA2G10 (*n* = 7) or 200 nM H48-PLA2G10 (*n* = 7). n represents the number of biological replicates. The statistical difference in the mean was assessed using *t*-test. *p*-value, as indicated. NS, not significantly different.

**Figure 4 ijms-23-08033-f004:**
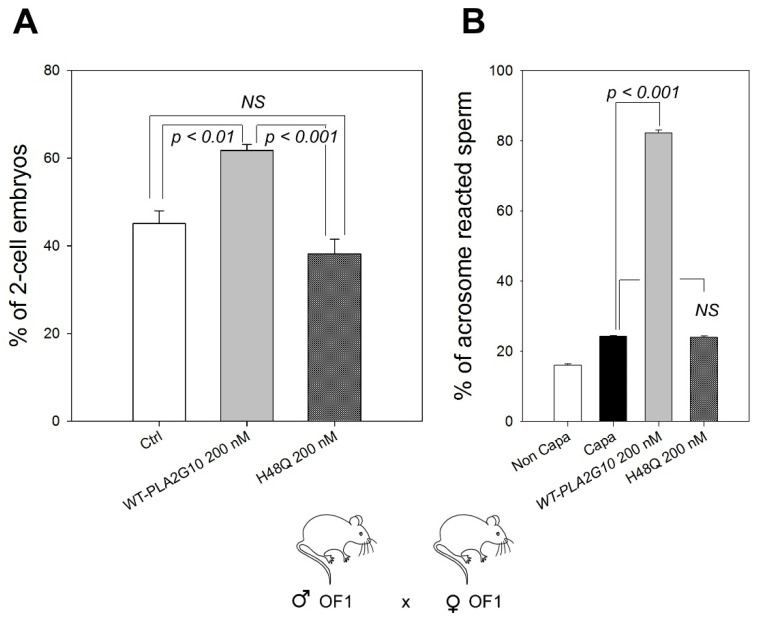
The H48Q-PLA2G10 mutant is unable to improve fertilization in the OF1 strain. (**A**) Percentages of two-cell embryos obtained with sperm and oocytes from OF1 animals. OF1 sperm were capacitated for a total duration of 55 min in M16-2% BSA and incubated at the end of the capacitation period with water (control, *n* = 6), WT-PLA2G10 (*n* = 6) or its catalytically-inactive mutant H48Q-PLA2G10 (*n* = 6). After treatment, washed sperm were introduced in droplets containing OF1 oocytes, and IVF outcomes were scored 24 h later. Statistical comparisons were made with a paired *t*-test. (**B**) Sperm were capacitated for 45 min in M16 medium containing 2% BSA, and capacitated sperm were treated for 10 min with either 200 nM of WT-PLA2G10 or H48Q-PLA2G10 and acrosome-reacted sperm were scored (*n* = 6). n represents the number of biological replicates, and for each replicate, more than 100 sperm cells were assessed per condition. The statistical difference in the mean was assessed using the *t*-test. *p*-value, as indicated. NS, not significantly different.

**Figure 5 ijms-23-08033-f005:**
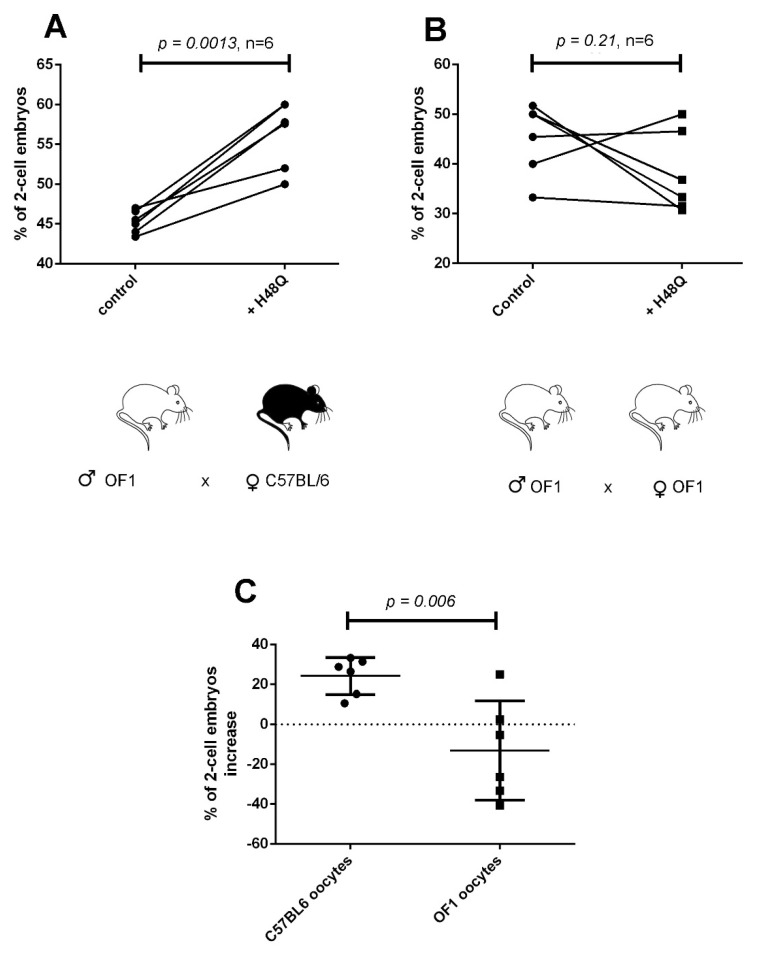
Impact of the strain origin of oocytes on H48Q-PLA2G10-induced improvement of IVF performed with OF1 sperm. (**A**) Percentages of two-cell embryos obtained with sperm from OF1 males and oocytes from C57BL/6 females. Sperm were untreated (control) or treated during capacitation with 200 nM H48Q-PLA2G10 as described above. (**B**) Percentages of two-cell embryos obtained with sperm and oocytes from OF1 males and females. Sperm were untreated (control) or treated during capacitation with H48Q-PLA2G10. (**C**) Comparison of the % of improvement of 2-cell embryos generated with sperm from OF1 males and oocytes from C57BL/6 or OF1 females as indicated. Statistical comparisons were made with paired *t*-test (**A**,**B**) or *t*-test (**C**).

**Figure 6 ijms-23-08033-f006:**
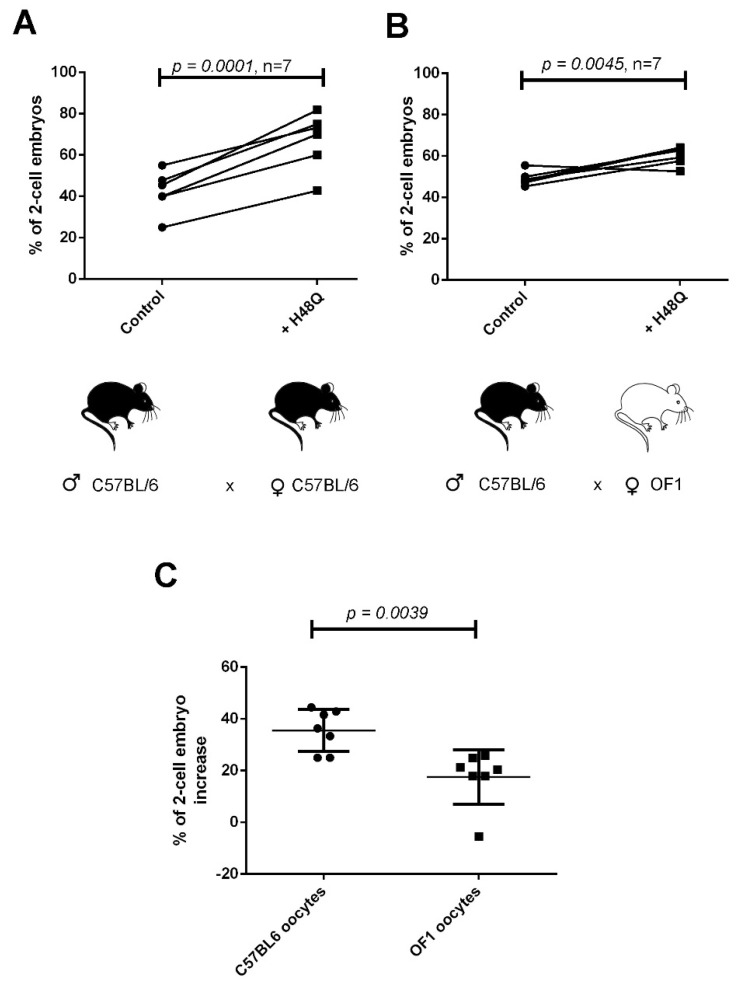
Impact of strain origin of oocytes on H48Q-PLA2G10-induced improvement of IVF performed with C57BL/6 sperm. (**A**) Percentages of 2-cell embryos obtained with sperm and oocytes from C57BL/6 males and females, respectively. Sperm were untreated (control) or treated during capacitation with 200 nM H48Q-PLA2G10, as described in Figure 3. (**B**) Percentages of two-cell embryos obtained with sperm from C57BL/6 males and oocytes from OF1 females. Sperm were untreated (control) or treated during capacitation with H48Q-PLA2G10. (**C**) Comparison of the % of improvement of 2-cell embryos generated with sperm from C57BL/6 males and oocytes from C57BL/6 or OF1 females as indicated. Statistical comparisons were made with paired *t*-test (**A**,**B**) or *t*-test (**C**).

**Figure 7 ijms-23-08033-f007:**
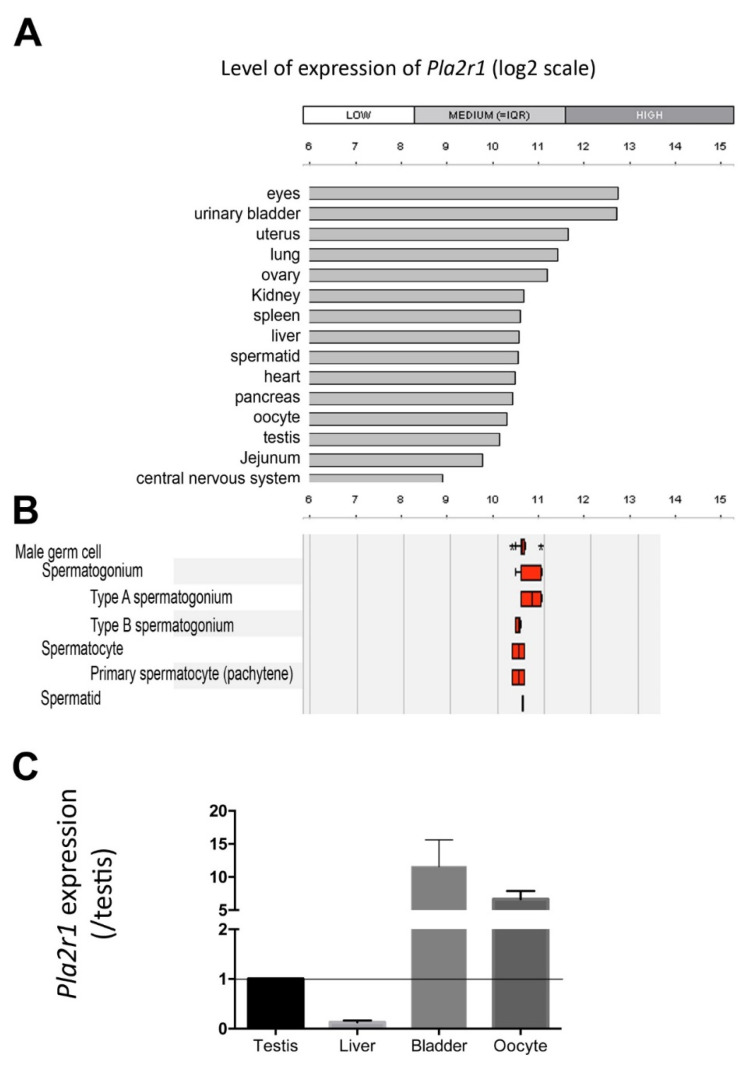
Pla2r1 is expressed in mouse oocytes and sperm. (**A**) Comparison of expression values in mouse tissues obtained by Affymetrix studies and extracted from data compiled by Genevestigator TM (https://genevestigator.com/, accessed on 28 April 2021). (**B**) Expression values in mouse male germ cells obtained by Affymetrix studies and extracted from data compiled by Genevestigator TM. (**C**) Expression profile of *Pla2r1* in mouse liver, bladder and oocyte relative to its expression in testis. Expression was quantified by RT-qPCR with *Gapdh* as the reference control. Measurements were performed 4 times for the testis, liver, and bladder and 5 times for the oocytes.

**Figure 8 ijms-23-08033-f008:**
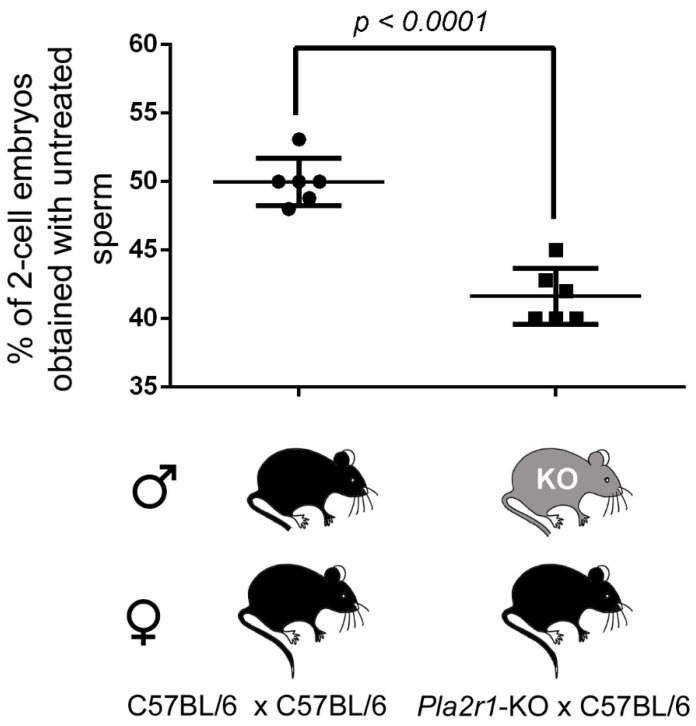
IVF outcome is decreased when sperm were obtained from *Pla2r1*-KO males. Comparison of the % of two-cell embryos generated by IVF, using oocytes from C57BL/6 and sperm from either WT or *Pla2r1*-KO littermate males (*n* = 6 for KO and WT males). n represents the number of biological replicates, and the statistical difference was assessed using the *t*-test and *p*-value, as indicated.

**Figure 9 ijms-23-08033-f009:**
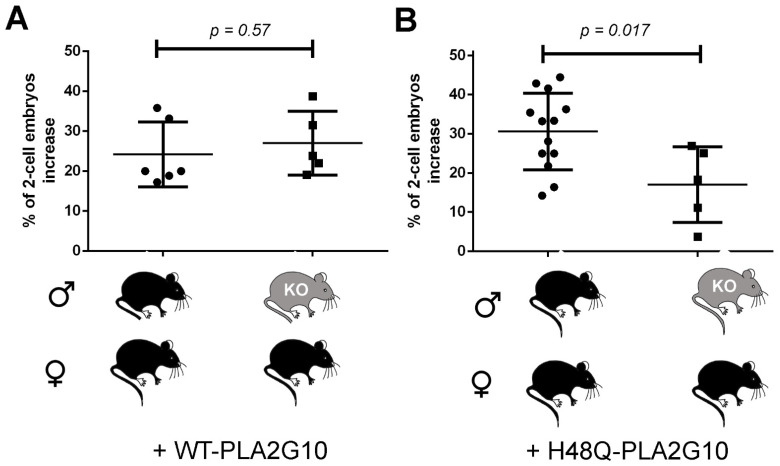
IVF efficiencies are similarly improved when sperm from C57BL/6 and *Pla2r1*-KO males are treated by WT-PLA2G10 in contrast to what is observed in sperm from C57BL/6 and *Pla2r1*-KO males treated by H48Q-PLA2G10. (**A**) Comparison of the % of improvement of 2-cell embryos generated with 200 nM WT-PLA2G10-treated sperm from C57BL/6 or *Pla2r1*-KO males and oocytes from C57BL/6 females as indicated. (**B**) Comparison of the % of improvement of 2-cell embryos generated with 200 nM H48Q-PLA2G10-treated sperm from C57BL/6 or *Pla2r1*-KO males and oocytes from C57BL/6 females, as indicated. Statistical comparisons were made with the *t*-test.

**Figure 10 ijms-23-08033-f010:**
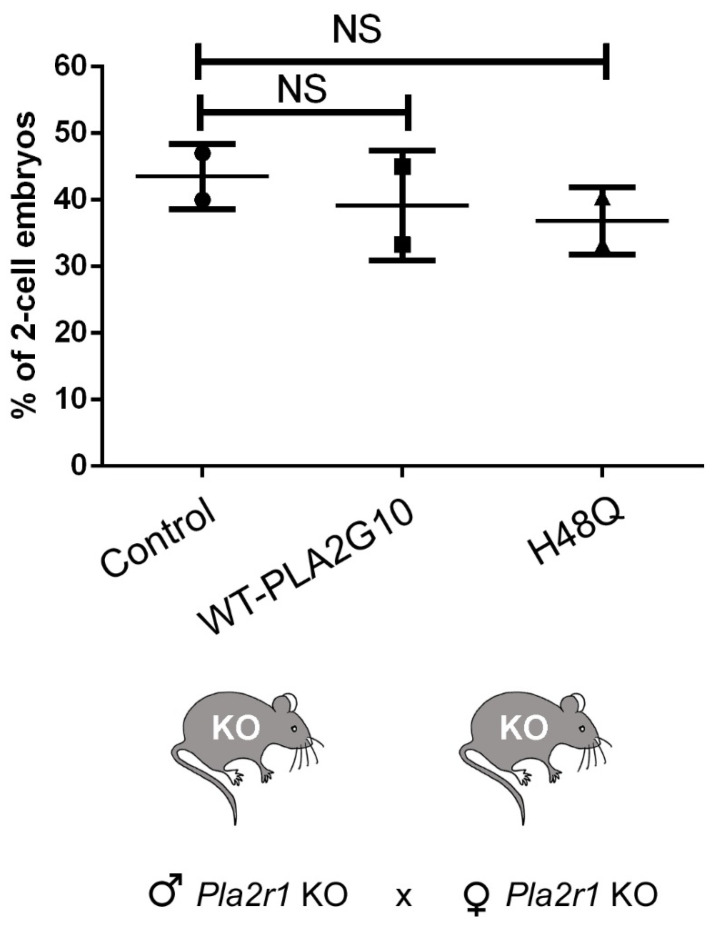
Improvement of fertilization by WT-PLA2G10 and H48Q-PLA2G10 is lost when IVF is performed with sperm and oocytes from *Pla2r1*-KO animals. Comparison of the % of two-cell embryos obtained by IVF using WT-PLA2G10 or H48Q-PLA2G10 treated sperm from *Pla2r1*-KO males and oocytes from *Pla2r1*-KO females (*n* = 2); no statistical difference (NS).

**Figure 11 ijms-23-08033-f011:**
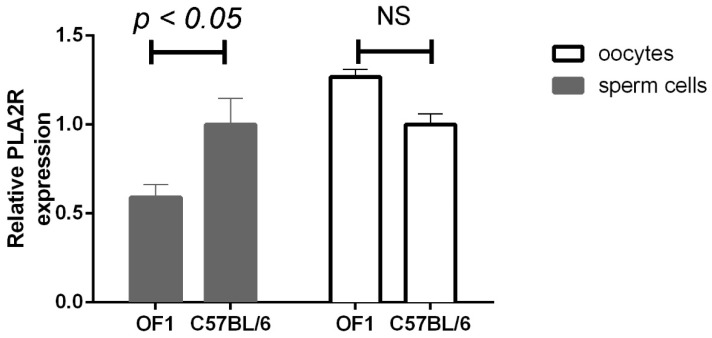
Comparative expression of *Pla2r1* in both gametes from OF1 and C57BL/6 strains. RNAs were extracted from 2 caudae epididymides and 50 oocytes at the germinal vesicle stage from OF1 and C57BL/6 males and females and their expression compared by RT-qPCR. Statistical comparisons were made with a paired *t*-test; no statistical difference (NS).

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
