# Peer review of "Treatment of Mouse Sperm with a Non-Catalytic Mutant of PLA2G10 Reveals That PLA2G10 Improves In Vitro Fertilization through Both Its Enzymatic Activity and as Ligand of PLA2R1"

_ijms, 2022, doi:10.3390/ijms23148033_

Round 1

Reviewer 1 Report

In this manuscript, the authors analyze if PLA2G10 participle in mice fertility as enzyme or as ligands for membrane receptors, using a catalytically-inactive mutant of PLA2G10 with low enzymatic activity but high binding properties to PLA2R1. The mutant did not trigger the AR but was as potent as WT-PLA2G10 to improve IVF in B6 mice but not in OF1 mice. Sperm from B6 Pla2r1-deficient mice were less fertile and lowered the profertility effects of H48Q-PLA2G10, which was completely suppressed when sperm and oocytes were collected from Pla2r1-deficient mice, but the effect of WT PLA2G10 was not sensitive to the absence of PLA2R1. The authors concluded that PLA2G10 acts as enzyme and ligand of PLA2R1.  The work is mainly based on IVF results, but the IVF protocol is poorly detailed, and what is described seems inadequate and inefficient.

The authors should clarify some points. In the first place, the H48Q mutant affects ivf only in B6 inbred mice, but not OF1, something difficult to understand, what is normal and what is the exception. It would seem that the exception is the inbred B6 strain, but further experiments on other strains should clarify this issue since it is essential to conclude the role of catalytic activity. Strains with clear differences in motility and sperm quality.

The results of IVF of Fig. 3 (control group) look very different of the result of the same inbred mice in Fig 8.

It would be important to know how many of the 2-cell embryos produced in the different treatments develop into blastocysts.

Indicate in M&M how long the males were individualized before IVF. Indicate if they made sperm selection to use only motile sperm.

The methodology they use to obtain the sperm is not the most appropriate for several reasons (Mouse sperm, obtained by manual trituration of caudae epididymides), trituration of the epididymis is not adequate, and it is better to use only sperm from the tail of the epididymis and the vas deferens.

It is surprising that the authors have not analyzed the role of the mutant H48Q in ko mice for Pla2g10.

The percentages of IVF obtained in the control group for B6 or OF1 in this work are very low (40%) compared to other works (65% DOI 10.1095/biolreprod.102.007344). It would be convenient to optimize the IVF protocol before analyzing any other factor.

Author Response

In this manuscript, the authors analyze if PLA2G10 participle in mice fertility as enzyme or as ligands for membrane receptors, using a catalytically-inactive mutant of PLA2G10 with low enzymatic activity but high binding properties to PLA2R1. The mutant did not trigger the AR but was as potent as WT-PLA2G10 to improve IVF in B6 mice but not in OF1 mice. Sperm from B6 Pla2r1-deficient mice were less fertile and lowered the profertility effects of H48Q-PLA2G10, which was completely suppressed when sperm and oocytes were collected from Pla2r1-deficient mice, but the effect of WT PLA2G10 was not sensitive to the absence of PLA2R1. The authors concluded that PLA2G10 acts as enzyme and ligand of PLA2R1.  The work is mainly based on IVF results, but the IVF protocol is poorly detailed, and what is described seems inadequate and inefficient.

The authors should clarify some points. In the first place, the H48Q mutant affects ivf only in B6 inbred mice, but not OF1, something difficult to understand, what is normal and what is the exception. It would seem that the exception is the inbred B6 strain, but further experiments on other strains should clarify this issue since it is essential to conclude the role of catalytic activity. Strains with clear differences in motility and sperm quality.

Numerous publications report notable and significant differences between different mouse strains. These differences can affect anatomy, receptor expression, pharmacological response to different agents. Since all strains of mice, both inbred and outbred, show differences from the wild-type strain, it is impossible to say whether it is the OF1 or C57BL/6 strain that shows a normal phenotype. We appreciate the suggestion to test other strains of mice and this will be the subject of future studies

The results of IVF of Fig. 3 (control group) look very different of the result of the same inbred mice in Fig 8.

First, it is not exactly the same inbred mice. It is well known that even C57BL/6 for two different commercial providers are slightly different. Second although the genetic background of PLA2R1 KO is C57BL/6, we can expect slight differences between C57BL/6 from Charles River and the KO. Nevertheless, there is no statistical difference between these two strains. We enclose for the reviewer the corresponding graph

It would be important to know how many of the 2-cell embryos produced in the different treatments develop into blastocysts.

Our study is restricted to the early steps of embryo development. Moreover, in OF1 mice, there is a two-cell block, preventing the study of embryo development further the 2-cell stage. Therefore, we did not measure this parameter in both strains.

Indicate in M&M how long the males were individualized before IVF. Indicate if they made sperm selection to use only motile sperm.

Males were maintained in individual cages for several weeks before their use. They have never been maintained with females before their use.

The methodology they use to obtain the sperm is not the most appropriate for several reasons (Mouse sperm, obtained by manual trituration of caudae epididymides), trituration of the epididymis is not adequate, and it is better to use only sperm from the tail of the epididymis and the vas deferens.

From my point of view, caudae epididymis and tail of the epididymis are similar. This protocol is used by numerous world recognized laboratories studying in vitro fertilization (see PMID: 27627854 for instance). My laboratory is studying mouse  in vitro fertilization since 1995 and no reviewers has made any remark about our IVF protocol. Never.

It is surprising that the authors have not analyzed the role of the mutant H48Q in ko mice for Pla2g10.

It is done in figure 9B and figure 10

The percentages of IVF obtained in the control group for B6 or OF1 in this work are very low (40%) compared to other works (65% DOI 10.1095/biolreprod.102.007344). It would be convenient to optimize the IVF protocol before analyzing any other factor.

There is a great variability in the % of 2-cell embryos obtained by IVF with C57BL/6 gametes. For instance, in PMID: 27627854, from Visconti’s lab, the mean percentage of 2-cell embryos is 45%; a value in agreement with ours.

Reviewer 2 Report

In this paper, Abi-Nahed and colleagues analyzed the role of phospholipase A2 (PLA2G10) during mouse fertilization. 

The paper is well conducted, and it may be of interest for those working in the field, for this, I recommend for its publication in IJMS prior these minor revisions: 

-       The Abstract and the main text lack a conclusion, summarizing the main findings of the paper

-       What do the authors intend with “unique enzymatic activity”, as stated in line 41?

-       It is not clear the temporal action of PLA2G10, indeed they assessed that “PLA2G10 is located in the acrosome and secreted during the acrosome reaction. At the cellular level, the enzyme participates to sperm membrane lipid changes induced by capacitation, a maturation step necessary for the acrosome reaction”; so, PLA2G10 is secreted during acrosome reaction, or stimulates acrosome reaction, changing the membrane lipid content? Please, clarify

-       It would be of helpful for the readers to add a picture of stained sperm with Coomassie blue, also explaining the criteria used for acrosome reaction classification (i.e., does this technique allows to distinguish between reacted sperm from those possessing abnormal acrosome, that may appear as “reacted”?)

Author Response

In this paper, Abi-Nahed and colleagues analyzed the role of phospholipase A2 (PLA2G10) during mouse fertilization. 

The paper is well conducted, and it may be of interest for those working in the field, for this, I recommend for its publication in IJMS prior these minor revisions: 

-       The Abstract and the main text lack a conclusion, summarizing the main findings of the paper

Done. We have added this sentence at the end of the abstract

This study shows that the action of PLA2G10 on gametes is complex and can simultaneously activate the catalytic pathway and the PLA2R1-dependent receptor pathway. This work also shows for the first time that PLA2G10 binding to gametes' PLA2R1 participates in fertilization optimization

-       What do the authors intend with “unique enzymatic activity”, as stated in line 41?

We have modified the sentence and added a reference

[2], the group X sPLA2 (PLA2G10) has unique enzymatic properties, with high affinity for arachidonic acid containing phospholipids [3]  

-       It is not clear the temporal action of PLA2G10, indeed they assessed that “PLA2G10 is located in the acrosome and secreted during the acrosome reaction. At the cellular level, the enzyme participates to sperm membrane lipid changes induced by capacitation, a maturation step necessary for the acrosome reaction”; so, PLA2G10 is secreted during acrosome reaction, or stimulates acrosome reaction, changing the membrane lipid content? Please, clarify

 That's a very good point. It is indeed admitted that the acrosome reaction can only take place when the spermatozoon has been capacitated. However, in vitro, during capacitation (performed with a high concentration of BSA), a part of the sperm cells performs a spontaneous acrosome reaction and releases enzymes into the incubation medium, thus modifying the other capacitating sperm. Thus, at the population level, the early acrosomal reaction of some sperm impacts the lipid modifications occurring during capacitation and overall, the lipid composition of WT sperm and of PLA2G10 KO males are different.

We have modified the sentence and cited a reference

[5]. During in vitro mouse sperm capacitation, a maturation step necessary for the acrosome reaction, a significant proportion of sperm performs spontaneous early acrosome reaction [5], and enzymes released from these sperm modify the lipid composition of capacitating sperm. Thus at

-       It would be of helpful for the readers to add a picture of stained sperm with Coomassie blue, also explaining the criteria used for acrosome reaction classification (i.e., does this technique allows to distinguish between reacted sperm from those possessing abnormal acrosome, that may appear as “reacted”?)

We have modified figure 1 according to reviewer’s suggestion.

Round 2

Reviewer 1 Report

It has been known for 20 years that the OF1 strain does not suffer from developmental blockage if a suitable culture medium, for example KSOM, is used.

A group that works on IVF in mice cannot obtain a 40% cleavage as a control, which is much lower than expected, unless they want to see significant differences with the experimental groups that they are analyzing. Part of the 2-cell embryos may be parthenotes or fragmentation, and may not have been fertilized.